# Landscape composition and local floral resources influence foraging behavior but not the size of *Bombus impatiens* Cresson (Hymenoptera: Apidae) workers

Amélie Gervais[1]*, Ève Courtois[2], Valérie Fournier[1], Marc Bélisle[2]

**1** Département de Phytologie, Centre de Recherche et d'Innovation sur les Végétaux (CRIV), Université Laval, Quebec City, Canada, **2** Département de Biologie, Centre d'Étude de la Forêt (CEF), Université de Sherbrooke, Sherbrooke, Québec, Canada

* amelie.gervais.bio@gmail.com

**Data Availability Statement:** All data and script files are available from the Dryad database. DOI: 10.5061/dryad.fxpnvx0ns.

## Abstract

Bumble bee communities are strongly disrupted worldwide through the population decline of many species; a phenomenon that has been generally attributed to landscape modification, pesticide use, pathogens, and climate change. The mechanisms by which these causes act on bumble bee colonies are, however, likely to be complex and to involve many levels of organization spanning from the community down to the least understood individual level. Here, we assessed how the morphology, weight and foraging behavior of individual workers are affected by their surrounding landscape. We hypothesized that colonies established in landscapes showing high cover of intensive crops and low cover of flowering crops, as well as low amounts of local floral resources, would produce smaller workers, which would perform fewer foraging trips and collect pollen loads less constant in species composition. We tested these predictions with 80 colonies of commercially reared *Bombus impatiens* Cresson placed in 20 landscapes spanning a gradient of agricultural intensification in southern Québec, Canada. We estimated weekly rate at which workers entered and exited colonies and captured eight workers per colony over a period of 14 weeks during the spring and summer of 2016. Captured workers had their wing, thorax, head, tibia, and dry weight measured, as well as their pollen load extracted and identified to the lowest possible taxonomic level. We did not detect any effect of landscape habitat composition on worker morphology or body weight, but found that foraging activity decreased with intensive crops. Moreover, higher diversity of local floral resources led to lower pollen constancy in intensively cultivated landscapes. Finally, we found a negative correlation between the size of workers and the diversity of their pollen load. Our results provide additional evidence that conservation actions regarding pollinators in arable landscapes should be made at the landscape rather than at the farm level.

**Funding:** MB: NSERC, Discovery Grand Program. https://www.nserc-crsng.gc.ca/professors-professeurs/grants-subs/dgigp-psigp_eng.asp VF: Centre-Sève, New Initiatives Program-Interinstitutional project. https://centreseve.recherche.usherbrooke.ca/en/node/205 The funders had no role in study design, data collection and analysis, decision to publish, or preparation of the manuscript.

**Competing interests:** The authors have declared that no competing interests exist.

## Introduction

Bumble bees are economically important pollinators worldwide. Their morphology and behavior make them sturdy pollinators, able to forage in colder and wetter conditions than most pollinating insects [1]. Through buzz pollination, they excel at pollinating a variety of crops such as tomatoes [2], cranberries [3], blueberries [4], apples [5], and haskaps [6]. Moreover, some species can be reared like honey bees for commercial purposes [7] to pollinate crops in greenhouses [2] or in fields [8]. Currently, there are ~ 250 known species worldwide, most of them found in temperate, alpine or arctic zones [9].

The global conservation status of bumble bees is unfortunately alarming, as not only populations but entire species are experiencing steep declines or even facing extinctions. For instance, Cameron et al. [10] found that the relative abundance of four bumble bee species has declined by up to 96% within the last 20 years in the United States. Besides, the limits of the southern range of most of the 67 species investigated by Kerr et al. [11] have shrunk across continents within the last 110 years. These falloffs do not appear to be random given that some species are more at risk than others [12,13]. Scientists attribute these declines to a combination of factors, including climate change [11], intensive use of pesticides [14–18], diseases, parasites and predation [19–21], as well as landscape simplification [22,23] and the subsequent decrease in local floral resource availability [21,24,25].

Landscape habitat composition has been found to impact bumble bees at different levels of organization, likely through mechanisms that alter both the amount and quality of nesting and foraging habitats. Indeed, bumble bees have been found to show a trait-dependent vulnerability to landscape simplification characterized by larger fields of fewer crop types, simplified crop rotations and less marginal, non-crop habitats [23,26]. Species forming smaller colonies, building above-ground nests, and having long life-cycles and late queen emergence, were less abundant in more simplified landscape [23]. The amount of high-value foraging habitats was also found to positively impact the abundance of colonies and the survival of family lineages [22,27–29]. Furthermore, colony growth and reproduction are also superior in landscapes showing higher proportions of natural [24,30] or suburban areas [31,32]. Even at the individual level, workers found in simplified landscapes were smaller than those inhabiting more complex landscapes [33]. Landscape habitat composition thus seems to impact bumble bees at different scales of organization, spanning from communities down to populations, colonies and even individuals.

Empirical evidence regarding the influence of landscape configuration on bumble bees is divergent partly because of the diversity of configuration metrics used across studies. For example, landscape connectivity was not found to impact performance (mass and reproduction) of colonies [34]. Field size, a proxy of landscape simplification, was found to reduce the survival of family lineages [27] and the body size of workers [33]. Also, bumble bee abundance was found to increase with edge density, while species richness decreased with mean area of forest patches [35]. Bumble bees foraging in fragmented landscape visited more flowers, flew longer total distances and tended to stay longer in fragments, compared to those foraging in unfragmented landscape [36]. The identification of overarching trends from previous studies is impeded by several other limitations. First and foremost, the lack of a clear theoretical framework regarding how landscape structure should affect the ecology of organisms [37]. Second, landscape configuration depends partly on landscape habitat composition. Third, studies vary simultaneously in the metrics and spatial scales used to quantify landscape configuration, as well as in the species and levels of organization targeted (i.e., community, population, colony or individual).

Key aspects related to landscape structure, usually investigated outside this concept given that they require measurements of greater resolution, include the effect of abundance and quality of floral resources and their spatio-temporal distribution on bumble bees. Virtually all studies that have examined the impact of floral resources (i.e. diversity, species richness, quality and/or abundance) have found a positive relationship between them and colony performance as measured through gyne and worker production, as well as colony mass [24,31,34,38–41]. These aspects are clear determinants of the fitness of colonies, but their effects could operate through individuals, notably workers for which ecological information remains sparse despite their pivotal role. Floral resource availability may not only dictate the number, size and morphology of workers that a colony should produce to maximise its fitness, but also the foraging behavior of workers [42–44]. In fact, workers of bumble bee nests placed next to mass-flowering crops, like *Phacelia tranacetifolia* Benth., made foraging trips of lower duration than those of nests placed elsewhere [44]. Just as trip duration, trip distance was reduced when colonies were placed in landscapes with more floral resources [45]. Given this, it would be reasonable to think that workers surrounded by a locally food-rich landscape should carry pollen loads that reflect the species composition of their local plant community, all other things being equal. Conversely, workers inhabiting locally poor landscapes likely need to forage farther and may thus encounter and carry pollen from plant species different from those found near their colonies. Such relationships have rarely been addressed [42] and could be complicated by the fact that flower selection is also influenced by floral resources. Workers indeed specialize in a few plant species (i.e., show flower constancy) when floral resources are abundant, which makes them more efficient at extracting pollen [46] (but see [47]). On the other hand, in poor floral habitats, workers tend to broaden their use of flower species likely as a way to reduce search costs for pollen sources [43,48–50].

Empirical evidence suggests that colony performance also increases with worker body size [51] and that larger workers outperform smaller ones at various tasks, including foraging [52,53]. Thus, colonies comprising mainly large workers will presumably acquire more resources and produce more workers and gynes than colonies with smaller workers [33,47,54–56]. Yet, larger workers are likely more costly to produce as their size seems to be directly linked to the quantity of food ingested during the larval stage [57,58]. The body size of workers should therefore be influenced by the landscape in which they were raised, especially in the first generations. To our knowledge, the only study that assessed the impact of landscape structure on the body size of bumble bee workers found that individuals captured in simplified landscapes were smaller [33]. Hence, colonies founded in landscapes characterized by a low floral resource availability may end up trapped in a vicious cycle, with small workers bringing less or lower quality food that would constrain the body size of the next generation of workers, a phenomenon that would ultimately result in the colony producing fewer or lower quality gynes.

Here, we quantify the influence of landscape composition and configuration, as well as of local floral resources, on 1) the body size and morphology, 2) the rate of entries and exits at the colony, and 3) the pollen loads of *Bombus impatiens* (Cresson) foraging workers. We also assess the correlation between worker size and pollen load diversity. We predicted that colonies experimentally placed in landscapes with a high proportion of intensively cultivated row crops would produce smaller workers with pollen loads more diverse and less similar to the local plant community, while those placed in landscapes with a high proportion of flowering crops and local floral resources would produce larger workers with pollen load less diverse and more similar to the local plant community. Finally, we hypothesized that larger workers would be more flower constant, which would result in more homogenous pollen loads than those collected by smaller workers.

## Methods

### Study sites

The study was carried out in Montérégie and Estrie, two regions of southern Québec (Canada). The study area (45˚12'45"- 45˚57'53"N; 71˚24'70"–73˚23'38"W; S1 Fig) is characterized by an east-west gradient of agricultural intensification. The eastern part is home to dairy and small-scale farms largely composed of hayfields and pastures, which are gradually replaced by large-scale, continuous, and intense row cropping farmlands mostly devoted to maize, soybean, and wheat to the west [59]. Forests follow a similar gradient whereby large expanses of forest in the east become gradually fragmented with increasingly smaller and more isolated forest patches in the west [60]. Also following the same gradient is the level of surface water contamination by pesticides, notably glyphosate and atrazine herbicides as well as several neonicotinoid insecticides, found in the streams and rivers of the western part of the study area [61,62].

### Colonies

A total of 20 commercial quads (Koppert Biological Systems©), each containing 4 colonies of *Bombus impatiens* (Cresson), were installed on 20 different farms on 3 May 2016, and monitored once a week until the natural death of colonies. Commercial colonies were provided with a vacuum sealed bag holding supplemental sugar water solution to help them survive through early spring conditions and give time for workers to learn their new spatial environment. Hence, the responses of bumble bees we measured should not be masked by habitat-specific adaptations and merely result from phenotypic plasticity. Farms were randomly selected from a haphazard sample of 40 interested producers with the constraint that they had to be spaced by > 5 km from each other (mean nearest-neighbor distance ± SD: 69 ± 37 km). The 20 farms covered a large portion of the potential range of combinations between the relative amount of extensive, flowering and intensive cultures that could be found within a 1-km radius around a given farm of our study area.

### Workers

Once a week, worker entries and exits from each colony were counted for 15 min. Counts were made between 9h00 and 16h00 and colonies were visited at different times across weeks to avoid confounding time effects. Starting two weeks after the colony was placed in the field, we captured one returning worker per colony per week whenever possible. Since most colonies survived up to 10 weeks, a maximum of 8 workers per colony was captured throughout the growing season mainly to ensure that colony performance would not be affected by the loss. We simply used a 50-ml Falcon™ conical centrifuge tube to capture workers. Workers were then put on ice and frozen at -20˚C upon return to the lab until further processing. After having removed their pollen loads (see Pollen sub-section below), we preserved sampled workers in 70% ethanol until morphometric measurement.

We measured several functional traits key to the role of workers: thorax dorsal (intertegular) width, wing length, marginal cell length, width and length of the head, and length of the tibia (pollen corbicula). Thorax dorsal width is positively correlated with the size of flight muscles and consists in the most common measure of bumble bee size [63]. The larger the thorax, the further a bumble bee can fly [64,65]. In the same vein, wing length, or wingspan, is generally positively associated with flight range in many pterygote insects [66], including bumble bees [67]. We also measured the width and length of the head, which are related to feeding habits [68]. Bumble bees with wider heads can be limited by the type of corolla they are able to access, while those with longer heads can feed on flowers with deeper corollas [69]. Finally, we

measured tibia length because a larger pollen corbicula is likely indicative of a greater capacity to collect pollen and bring larger loads back to the colony. Measurements (± 0.001 mm) were taken with an electronic Olympus SZX7 microscope fitted with a numeric Olympus IX-TVAD camera and its associated software (Olympus *stream basic*). Each trait was measured three times to ensure repeatability (max 0.1 mm between any two measures). Measurements were then averaged for all analyses. Lastly, we dried sampled bumble bees at 50˚C for 4 days before weighing (Adam Equipment, model AAA250L, ± 0.0001 g) individuals with no missing body parts.

## Pollen loads

We removed and colored the pollen carried by workers following Moisan-Deserres et al. [4]. We used repeated 70% alcohol rinses and a mounting needle over a 50-ml Falcon™ conical centrifuge tube to remove all the pollen found on a worker's body. Tubes were then centrifuged at 3,000 rpm for 12 min to isolate the pollen. The supernatant was then retrieved and the remaining liquid was left to evaporate overnight. We added 5 ml of tween 0.05% to the dry pollen and 20 µl of the solution was then put on a slide and colored using fuchsin-stained glycerin gelatin [70]. For each slide, a haphazard sample of 150 pollen grains was finally identified to the lowest taxonomic level possible by a melissopalynology expert equipped with microscope (1000x), namely Mélissa Girard [71,72].

## Landscape characterization

We conducted ground surveys to characterize the habitat composition of the landscapes surrounding experimental sites once during the summer (in August). All polygons (fields, roads, forests) delineated using orthophotos (scale, 1:15 000) were identified within a 1000-m radius of each colony. We considered a radius of 1000 m because it should include most of the bumble bees' foraging ranges [73]. We then calculated the proportion of land covered by the following land use types with QGis [74]: forest, water, urban area, intensive crops (i.e. crops generally treated with significant quantities of pesticides, such as corn, soybean, wheat and other small cereals, apples, strawberries), extensive crops (e.g. hay, pastures) and flowering crops (i.e. nectariferous crops not requiring substantial amounts of pesticides, if any, such as alfalfa, clover) (S2 Fig). This categorization was assumed to reflect the vegetation structure, the potential at providing nectar and pollen, and the contamination level by pesticides of the different land use types. We also considered the length of habitat margins (edges) within a 1000-m radius of each colony. Every ecotone delimiting any two land use types we used was considered and total margin length was computed by QGIS. This landscape metric was used since a vast array of floral resources of interest to pollinators such as bumble bees can generally be found within these margins and thus affect food resource availability [9]. Furthermore, margin length is often positively associated with agricultural landscape structural complexity, which is deemed favorable on many aspects to farmland biodiversity [26].

## Local floral resources

On each weekly visit, we identified to the lowest taxonomic level possible all blooming plants within a 100-m radius from colonies, and this, in order to estimate the weekly species richness of flower-bearing plants easily available to bumble bees (i.e., with low travel costs). For the analyses we used the mean species richness found locally within 10 weeks (number of weeks where all the colonies were still alive). Although we could not estimate the amount of nectar and pollen that each plant species could provide to bumble bees, something rather difficult to do in the field, our index is nevertheless in line with the recommendations of Szigeti et al. [75]

regarding field measurements of food resource availability based on visual units reflecting the pollinators' perspective.

## Statistical analyses

**Model selection.** We first ran models assessing the influence of landscape variables on each response variable (morphology, foraging activity, pollen load richness and habitat/pollen similarity) at each of two spatial scales, 500 and 1000 m, in order to determine at which of these two scales landscape structure was more likely to affect bumble bee workers. We made this comparison based on the second-order Akaike information criterion (AICc; [76]) using the most complex models we deemed justifiable to fit (see below). This led us to exclude models with landscape variables measured at 1000 m because those based on the 500-m scale performed systematically better, except for the thorax (mean ΔAIC = 0.29; S1 Table). Note that we restricted analyses regarding the morphology of workers to individuals captured at least four weeks after their colony was placed in the field to ensure that the potential effects on morphometrics would be associated to landscape variables, not the commercial rearing process. We standardized (zero mean and unit variance) all explanatory variables prior to analyses. We also assessed the level of collinearity among all explanatory variables used in models based on Pearson's correlation coefficients and variance inflation factors (S2 Fig). Landcover variables being compositional variables in the sense that they sum to a constant and are implicitly correlated among one another, it was not surprising that we had to exclude some of these from our analyses. This was the case of the proportion of forest as well as total margin length since they were collinear with the proportion of intensive crops, a central variable to our main research questions. Given that compositional variables also reduce the number of true alternative hypotheses that can be considered through a modeling exercise, we further excluded the proportion of extensive crops. We considered this landcover to be more neutral than others with respect to bumble bees as hayfields and pastures are rarely treated with pesticides and are characterized by a much lower food resource availability than alfalfa and clover fields (flowering crops), for example. Given this, it was difficult to make clear predictions as to the potential effects of extensive crops. Lastly, we excluded the proportion of urban area since it represented a low proportion of the area surrounding colonies and was not the focus of this research.

Lastly, we compared the performance of five candidate models for each response variable based on AICc (Table 1). The set of candidate models first included a null model (intercept only or intercept + confounders) in order to determine if landscape structure had any bearing on response variables. The second and third model assessed the influence of intensive and flowering crops and the additional information that could be provided by local flower species richness, respectively. The fourth model assessed the possibility that the influence of flowering crops can be modulated by the amount of intensive crops in the landscape. The fifth and last model allowed us to determine whether the influence of either intensive or flowering crops could be additionally modulated by local flower species richness. Large scale effects may indeed end up being less important in locally rich environments. All mixed models (see below) were fitted with the lme4 package (v.1.1–21 [77]) in R (v.3.5.0; [78]). The glmmTMB package (v.0.2.3; [79]) was however used for the foraging activity models because these converged more easily with this package. Model predictions and coefficients were model-averaged using the AICcmodavg package (v.2.2–2; [80]) and are reported with their 95% unconditional confidence intervals [76]. Residuals from the most complex model (#5; Table 1) for each response variable were tested against spatial coordinates using Moran's Index to assess potential spatial autocorrelation problems using the DHARMa R packages (v.0.3.1; [81]). No autocorrelation was found for any of the response variables.

**Table 1. Candidate models considered in model selection and multimodel inference procedures.** Models were compared based on the second-order Akaike information criterion (AICc). K: number of model parameters; w: Akaike weight. Bold characters represent the best model for each response variable. INT: Proportion of intensive crops; FLO: Proportion of flowering crops; LOC: Local floral richness; JJ: Julian day.

| Candidate models* | Weight | | | Thorax width | | | Wing length | | | Head length | | | Tibia length | | | Foraging activity | | | Pollen load richness | | | Difference pollen vs habitat | | |
|---|---|---|---|---|---|---|---|---|---|---|---|---|---|---|---|---|---|---|---|---|---|---|---|---|
| | K | ΔAICc | w | K | ΔAICc | w | K | ΔAICc | w | K | ΔAICc | w | K | ΔAICc | w | K | ΔAICc | w | K | ΔAICc | w | K | ΔAICc | w |
| 1 ~ Null | 4 | **0.00** | **0.76** | 4 | **0.00** | **0.82** | 4 | 1.85 | 0.20 | 4 | **0.00** | **0.80** | 4 | **0.00** | **0.79** | 5 | 198.72 | 0.00 | 4 | 1.92 | 0.17 | 2 | 21.75 | 0.00 |
| 2 ~ INT + FLO | 6 | 3.38 | 0.14 | 6 | 3.76 | 0.12 | 6 | **0.00** | **0.49** | 6 | 3.51 | 0.14 | 6 | 3.36 | 0.15 | 10 | 2.38 | 0.20 | 6 | **0.00** | **0.45** | 4 | 14.74 | 0.00 |
| 3 ~ INT + FLO + LOC | 7 | 5.57 | 0.05 | 7 | 5.90 | 0.04 | 7 | 1.94 | 0.19 | 7 | 5.68 | 0.05 | 7 | 5.54 | 0.05 | 11 | 3.86 | 0.10 | 7 | 1.34 | 0.23 | 5 | 16.43 | 0.00 |
| 4 ~ INT + FLO + LOC + INT:FLO | 8 | 5.46 | 0.05 | 8 | 8.09 | 0.01 | 8 | 3.31 | 0.09 | 8 | 7.84 | 0.02 | 8 | 7.75 | 0.02 | 12 | 5.86 | 0.04 | 8 | 3.36 | 0.08 | 6 | **0.00** | **1** |
| 5 ~ INT + FLO + LOC + INT:FLO + INT:LOC + FLO:LOC + INT:JJ | 11 | 8.91 | 0.01 | 11 | 14.71 | 0.00 | 11 | 8.91 | 0.01 | 11 | 11.59 | 0.00 | 11 | 12.59 | 0.00 | 14 | 11.02 | 0.00 | 11 | 4.03 | 0.06 | | | |
| 6 ~ Temperature + Time of the day | | | | | | | | | | | | | | | | 8 | **0.00** | **0.66** | | | | | | |

*Temperature and time of day were also included in all models (#1 to #5) for the foraging activity response variable. The interaction INT:JJ was not included in model #5 for the foraging activity response variable.

**Morphology.**   We modeled the influence of landscape variables on the morphology (thorax, wing, head, tibia) and dry weight of bumble bee workers using linear mixed models. All models included Julian date as a potential confounding variable given that workers' size varies through the course of the season [82]. Site (quad) identity was included as a random factor in all models. We also considered the identity of the colony of origin as an additional random factor, here as well as for other response variables, but this prevented some models to converge. Colony identity was therefore left out of all models. Since some workers had missing body parts, sample size varied among response variables that were analysed. Besides, we only used sites where at least five workers were collected after the fourth week of the experiment. We also applied a principal component analysis (PCA) on standardized morphology and weight measures in order to determine whether we could distinguish different morphological types based on a combination of functional traits. The PCA was computed with the vegan package (v. 2.5–1; [83]) in R. Only the workers with no missing body parts and captured after the fourth week of the experiment were used in the PCA.

**Foraging activity.**   The influence of landscape variables on the foraging activity of bumble bee workers (total of entries and exits per 15 min) was determined with generalized linear mixed models including a Poisson distribution and log link function. Potential confounding variables included in all models comprised temperature (°C), time of day (24 hour-clock system) and Julian date. We used an observation-level random effect in addition to site (quad) and colony identity to control for overdispersion [84].

**Pollen.**   We modeled the influence of landscape variables on the number of plant species found in pollen loads carried by bumble bee workers using generalized linear mixed models with a Poisson distribution and log link function. Sites where less than 5 workers carrying a pollen load could be captured were excluded from the analyses. We again used site (quad) identity and observation-level random effects to take potential overdispersion into account [84]. We further determined if landscape variables affected the difference in flower species composition of pollen loads and that found within 100 m of colonies as estimated by dissimilarity coefficients (Hellinger distances) computed from presence-absence matrices of the plant species present in both the pollen loads and the local environment [85]. Dissimilarity coefficients were computed with the adespatial package (v.0.3–7; [86]) and then regressed on landscape variable with a linear model. Finally, we assessed whether the flower species richness of pollen loads was correlated to worker morphology using generalized linear mixed models with a Poisson distribution and log link function. One model was fitted for each morphological trait (standardized) as an explanatory variable and included the site and hive as random effects and the local floral richness (standardized) as an additional fixed effect.

## Results

A total of 264 workers from 76 colonies (20 quads) were captured, measured and had their pollen load removed during the experiment in 2016. Among them, 205 workers were also dried and weighed.

### Morphology

Bumble bee foraging workers showed little variation in size with coefficient of variation ranging between 5.9% and 8.80% (except for dry weight: 36.1%) across functional traits (mean ± SD): thorax (intertegular) width: 3.943 ± 0.318 mm; wing length: 10.076 ± 0.887 mm; head length: 3.115 ± 0.185 mm; head width: 2.389 ± 0.144; tibia: 3.981 ± 0.350 mm; dry weight: 0.0490 ± 0.0177 g. All linear trait measurements were highly correlated among one another

**Table 2. Model-averaged coefficients and their unconditional 95% confidence intervals.** Model-averaged coefficients were estimated by multimodel inference following model selection (see Table 1) and computed with standardized explanatory variables (zero mean and unit variance). Parameters for which their confidence interval does not overlap zero are shown in bold text. The number of sites (S) and of workers (W) used in analyses are shown where applicable.

| Explanatory variables | Morphology | | | | | Foraging behavior | | |
|---|---|---|---|---|---|---|---|---|
| | Thorax width (S: 19; W:176) | Wing length (S: 15; W:95) | Head length (S: 19; W; 177) | Tibia length (S: 19; W:165) | Weight (S: 19; W: 147) | Activity (S: 20) | Pollen load richness (S: 19; W: 201) | Difference pollen vs habitat (S: 20) |
| INT (models 2,3,4,5) | 0.01; [-0.05, 0.06] | -0.07; [-0.32, 0.18] | -0.01; [-0.05, 0.02] | -0.03; [-0.09, 0.03] | 0.00; [0.00,0.00] | **0.17; [-0.32, -0.01]** | -0.07; [-0.18, 0.05] | **0.02; [0.01, 0.07]** |
| FLO (models 2,3,4,5) | 0.00; [0.00, 0.00] | 0.00; [0.00, 0.00] | 0.00; [0.00, 0.00] | 0.00; [0.00, 0.00] | 0.00; [0.00,0.00] | 0.09; [-0.06, 0.25] | 0.00; [0.00, 0.00] | -0.01; [-0.02, 0.01] |
| LOC (models 3,4,5) | 0.00; [0.00, 0.00] | 0.00; [0.00, 0.00] | 0.00; [0.00, 0.00] | 0.00; [0.00, 0.00] | 0.00; [0.00,0.00] | 0.04; [-0.02, 0.01] | 0.00; [0.00, 0.00] | 0.01; [-0.01, 0.02] |
| INT X FLO (models 4,5) | 0.00; [-0.06, 0.07] | -0.16; [-0.50, 0.17] | 0.00; [-0.04, 0.04] | 0.00; [-0.07, 0.07] | 0.00; [0.00,0.00] | 0.03; [-0.14, 0.20] | 0.07; [-0.10, 0.25] | **-0.03; [-0.04, -0.02]** |
| FLO X LOC (model 5) | 0.00; [0.00, 0.00] | 0.18; [-0.12, 0.47] | -0.01; [-0.05, 0.03] | -0.04; [-0.11, 0.03] | 0.00; [0.00,0.00] | -0.02; [-0.09, 0.05] | 0.08; [-0.04, 0.20] | - |
| INT X LOC (model 5) | 0.00; [-0.09, 0.09] | -0.30; [-0.07, 0.10] | -0.04; [-0.09, 0.01] | -0.02; [-0.12, 0.08] | 0.00; [-0.01,0.00] | -0.01; [-0.10, 0.08] | **0.19; [0.03, 0.34]** | - |
| JJ (models 1,2,3,4,5,6) | -0.02; [-0.09,0. 05] | -0.02; [-0.36, 0.04] | 0.00; [-0.04, 0.04] | 0.04; [-0.03, 0.12] | 0.00; [0.00,0.00] | **-0.89; [-0.97, -0.81]** | **0.21; [0.11, 0.30]** | - |
| JJ X INT (model 5) | -0.01; [-0.09,0.06] | 0.16; [-0.31, 0.63] | -0.02; [-0.06, 0.02] | -0.03; [-0.11, 0.04] | 0.00; [0.00, 0.00] | - | 0.04; [-0.06,0.14] | - |
| Temp. (models 1–6) | - | - | - | - | - | -0.01; [-0.09, 0.07] | - | - |
| TOD (models 1–6) | - | - | - | - | - | 0.01; [-0.05, 0.08] | - | - |

(0.82 ≤ r ≤ 0.94) and with (log-transformed) dry weight (0.68 ≤ r ≤ 0.77). Since marginal cell length and head width were highly correlated with wing length ($r = 0.97$) and head length ($r = 0.93$), respectively, their relationships with landscape variables were not investigated to avoid duplication. None of the traits we considered was related to landscape habitat composition nor to local floral species richness (Table 2). In fact, the best model was the null for all traits (0.76 ≤ w ≤ 0.82), except for wing length ($w = 0.20$; Table 1). Although the second-best model was systematically the one that included simple effects of crop covers and local floral species richness across traits, Akaike weights remained relatively low (0.12 ≤ w ≤ 0.15), again with the exception of wing length ($w = 0.49$; Table 1). Lastly, we did not find different morphological types of bumble bee foraging workers as no distinct groups of observations arose in the unconstrained multivariate space formed by all functional traits (Fig 1), inasmuch as the total inertia (i.e., sum of the total variance) of the PCA was only 0.0002. The first two axes displayed 77.01% of the total variance.

## Foraging activity

The number of bumble bee workers entering and exiting a colony in 15 min was best explained by the model that only included time of day and temperature ($w = 0.66$) and the one that only included main effects of crop covers ($w = 0.20$; Table 1). Yet, we only found a negative effect of intensive crops on foraging worker activity (Table 2; Fig 2). Model selection thus provided no evidence that the effects of crop covers or local floral resources were modulated by one another. Finally, we found that foraging activity decreased over the course of the season (Table 2).

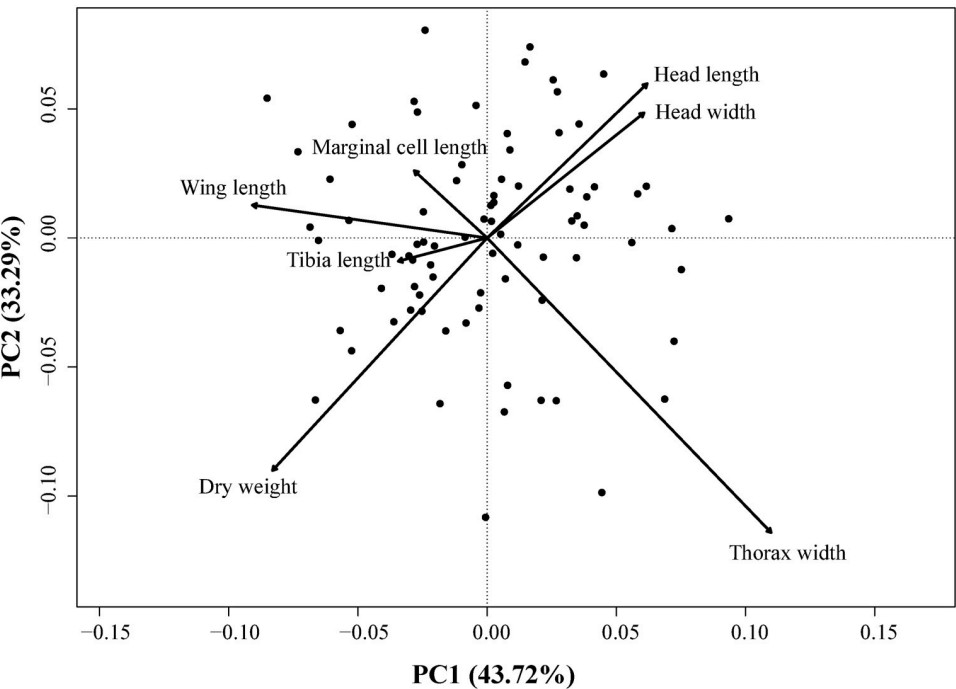

**Fig 1. Principal component analysis (PCA).** Plot showing the multivariate variation among 80 *Bombus impatiens* workers with respect to their morphometric measurements. Vectors indicate the direction of the functional trait to the overall distribution. The first two principal axes explained 77.01% of the variance.

## Pollen

Bumble bee workers collected pollen from 41 different species (morphotypes) of plants across the entire study area and over the entire study duration (S1 Table). While the species richness of pollen loads averaged 5.9 ± 1.7 across experimental sites, the number of species detected in pollen loads carried by individual workers had a mean of 5.8 ± 4.2 species. The species most frequently collected were *Picea* sp. (found on 56.6% of workers), *Taraxacum officinale* (60.5%), and *Salix* sp (62.9%). Species richness was best explained by the model that only included main effects of crop covers ($w = 0.45$), followed by the one that also included local floral species richness ($w = 0.23$; Table 1). Yet, we also found evidence that the effect of intensive crops was modulated by local floral resources (Table 2; Fig 3). Pollen load species richness decreased with intensive crop cover at low local floral species richness, but remained relatively stable when the latter was high (Fig 3).

The difference in plant species composition between pollen loads and the colony's surrounding (100-m radius) was best explained by the model including crop covers, local floral species richness, and the interaction between intensive and flowering crops ($w = 1.00$; Table 1). All of the other models were barely supported by the data (Table 1). Differences in species composition increased with intensive crops when the amount of flowering crops was low, but this effect decreased in size as the cover in flowering crops increased and was about null at high flowering crop covers (Table 2; Fig 4).

Finally, we looked at the correlation between the morphology of workers and the species richness of their pollen loads. The size of functional traits was negatively correlated with species richness, indicating that larger workers collected pollen loads composed of fewer plant species. When taking into account the local floral richness at each site, the functional traits

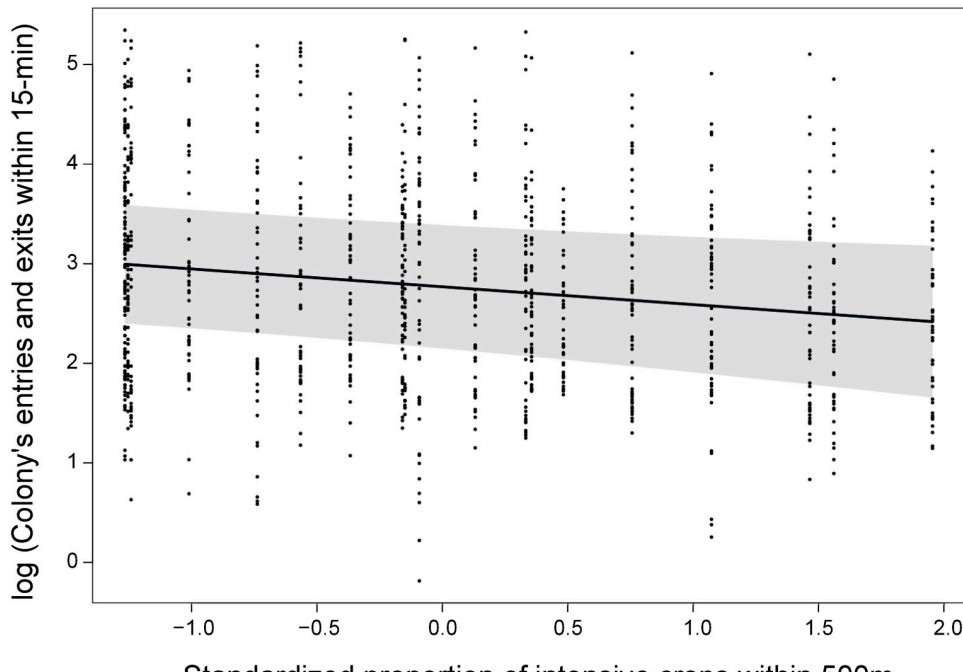

**Fig 2. Influence of intensive crops (500-m radius) on the foraging activity (log(entries and exits in 15 min)) of *Bombus impatiens* workers.** Data from the 20 clusters composed of 4 experimental colonies (quads) of *Bombus impatiens* monitored in 2016 in Southern Québec, Canada. Model-averaged predictions under average conditions are shown with 95% unconditional confidence intervals.

best correlated with pollen load species richness were wing length (slope estimate: -0.15; 95% CI [-0.21, -0.08]) and thorax width (-0.09; 95%CI [-0.15, -0.03]), and to a lesser extent, tibia length (-0.08; 95%CI [-0.14, -0.02]), marginal cell length (-0.07; 95%CI [-0.13, -0.01]), head length (-0.06; 95%CI [-0.12, 0.00]) and head width (-0.06; 95%CI [-0.12, 0.00]). Pollen load species richness was not found to be related to dry weight (-0.01; 95%CI [-0.13, 0.11]; Fig 5).

## Discussion

### Morphology

The morphology and size of *Bombus impatiens* workers were not influenced by either landscape habitat composition nor local floral species richness. This lack of relationship contradicts our predictions as well as the results from other research. Landscape simplification, habitat fragmentation and pesticide use were indeed previously found to lead to the production of smaller bumble bee workers and solitary bees [33,87–89]. Moreover, *B. impatiens* workers are known to show an important size difference (up to tenfold) even within a given colony [58]. This variation partly stems from the position of larvae in the nest, which in turn dictates the amount of food resources fed to future workers [58]. Greater food availability for the larvae also generally translates into larger workers [9,57,90]. Larger workers usually end up being foragers, and smaller ones manage the nest and raise the brood. In the event of a shortage of large foragers and deficient foraging, smaller workers should switch roles and begin to forage [9]. This is not the pattern we observed as foraging workers barely varied in morphology and body size, even among contrasting landscapes. One possible explanation for this could be that we used commercially reared bumble bee colonies, which were thus likely to be genetically similar.

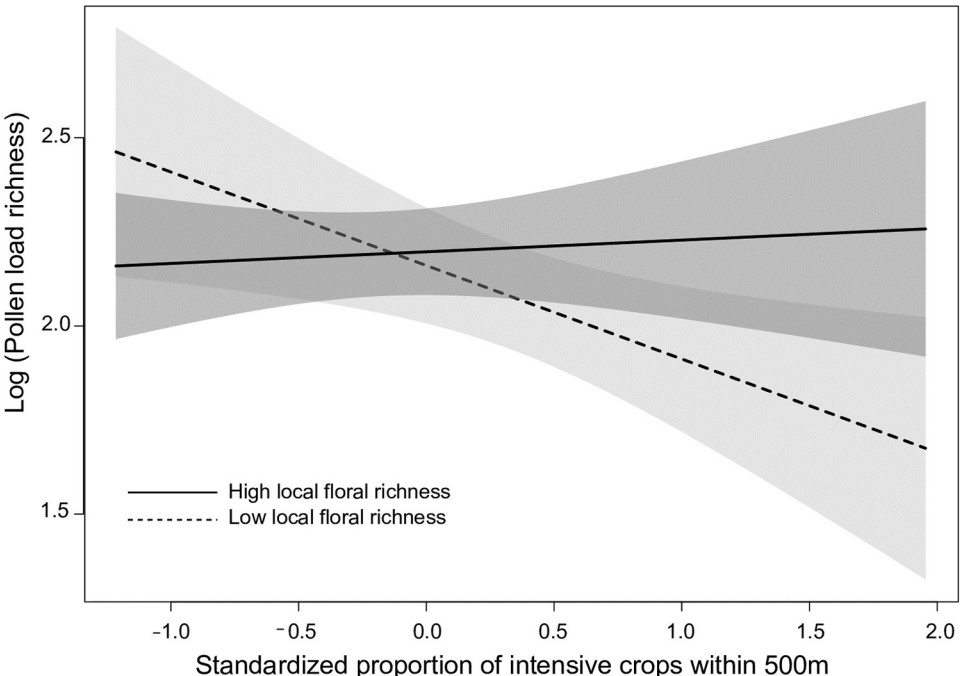

**Fig 3. Species richness found in pollen loads of 205 *Bombus impatiens* workers as a function of the interaction between the proportion of intensive crops (500-m radius) and local floral richness (100-m radius).** Workers were captured from 20 clusters composed of 4 experimental colonies of *Bombus impatiens* monitored in 2016 in Southern Québec, Canada. The dotted and solid lines represent pollen load richness when local floral richness was low (<8) and high (≥8 species), respectively. Model-averaged predictions under average conditions are shown with 95% unconditional confidence intervals.

A reduced genetic variance among colonies may have in turn induced lower levels of phenotypic variance or plasticity [91,92]. This being said, the decrease in worker body size of 4 bumble bee species over 125 years in New Hampshire, USA, as well as the fact that this decrease was more pronounced for the 2 largest species, which populations are also declining [93], certainly stress the need for further research regarding how landscape modification may affect bumble bee morphology and fitness, and ultimately population dynamics.

## Foraging activity

As predicted, the proportion of intensive crops had a negative impact on the foraging activity of bumble bee workers (Fig 2). Although studies addressing the influence of landscape structure on bumble bee foraging remain scarce, several studies have found that the occurrence of arable fields and pesticides in the landscape surrounding colonies can impact the foraging behavior and distance travelled by workers [45,94–98]. Evidence shows that pesticides, and notably neonicotinoid insecticides, can kill individuals or, when absorbed in sublethal doses, lead to disorientation, memory loss, and slower learning processes in bumble bees, honey bees, and stingless bees [17,94–96,99]. The negative effects of pesticides on bees, which are known to be weather- and landscape-dependent [100], may thus result in a less effective working force, and ultimately impact the foraging and fitness of colonies. Empirical support for these contentions is accumulating. For instance, Arce et al. [95] found that colonies exposed to clothianidin had lower return rates of foraging bees than control ones. Also, experimental bumble bee colonies established in intensively cultivated areas, generally show reduced weight

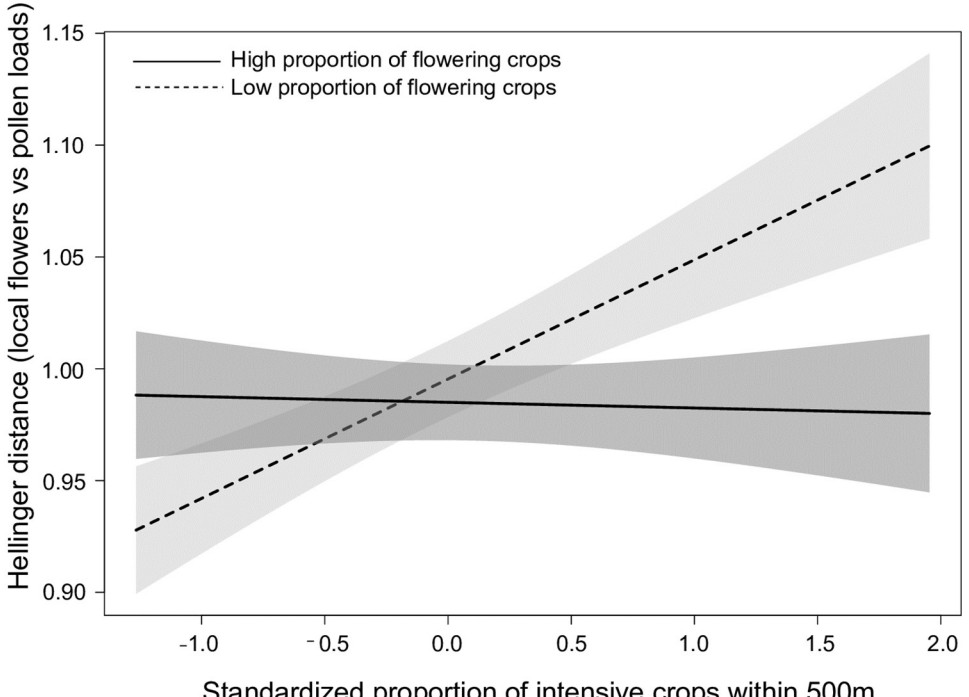

**Fig 4. Difference (Hellinger distance) in plant species composition of pollen loads of *Bombus impatiens* workers and surrounding habitat (100-m radius) as a function of the interaction between the proportion of intensive and flowering crops (500-m radius).** Workers were captured from 20 clusters composed of 4 experimental colonies of *Bombus impatiens* monitored in 2016 in Southern Québec, Canada. The dotted and solid lines represent cases where the proportion of flowering crops was low (<15%) and high (≥15%), respectively. Model-averaged predictions under average conditions are shown with 95% unconditional confidence intervals.

gains and survival rates [15,16,18,32,101]. Although we did not measure the levels of pesticide contamination in our experimental sites, the cover in intensive row crops has been found to be related to pesticide concentrations, including that of several neonicotinoids, in surface water [61,62] and insects [102] within our study area. Furthermore, the presence of arable fields in the landscape increased the foraging distance of all five bumble bee species studied by Redhead et al. [45], a consequence that may also lead to fewer and longer foraging trips [97], and ultimately contribute to less activity at the entrance of the colony.

We hypothesized that increasing amounts of flowering crops and local floral species richness would increase foraging activity by providing closer and more abundant food sources. In fact, Redhead et al. [45] found that a semi-natural habitat and flower density decreased the foraging distance of five bumble bee species, which should reflect positively on the activity level at the entrance of colonies, by encouraging workers to perform more frequent and shorter trips, [44]. Furthermore, mass flowering crops, such as oilseed, alfalfa and clover, were found to be beneficial to bumble bee colonies [103]. Yet, we found no effect of flowering crops or local plant richness on foraging activity. This could be explained by the fact that flowering crops are only in bloom for a shorter time than the period during which we monitored colonies and their workers' activity level. After blooming, flowering crops like alfalfa and clover could represent a relatively neutral habitat, where there is no harm aside from increasing distances, but also no food. Hence, flowering crops could have a positive, neutral or negative impact on the foraging activities of bumble bees depending on their own blooming phenology and that of alternative floral sources.

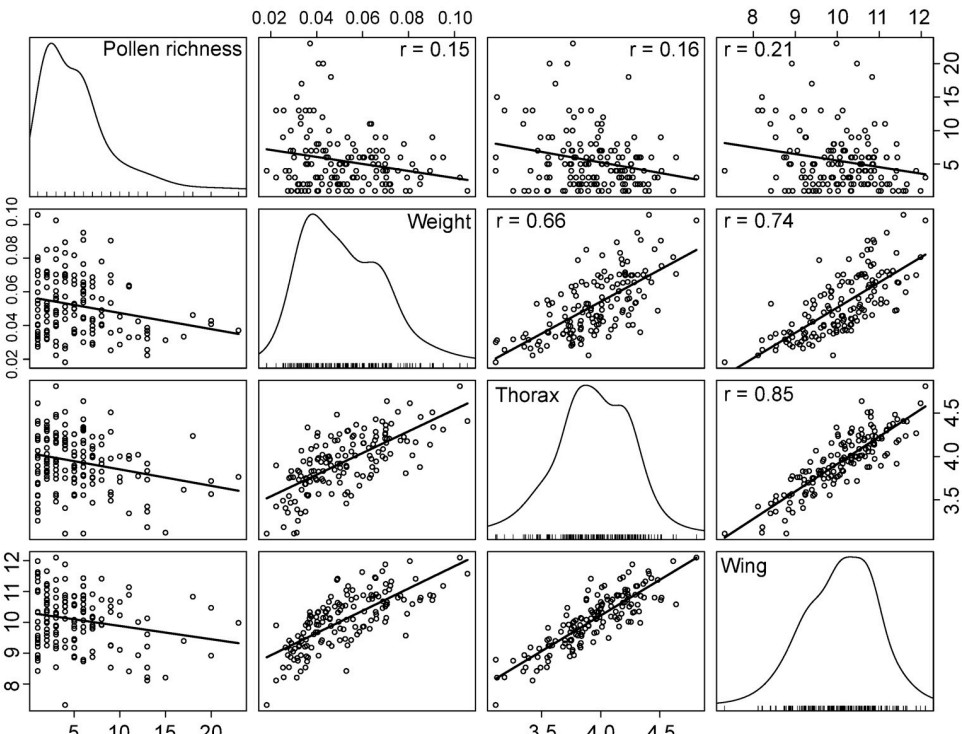

**Fig 5. Matrix of correlation between pollen richness, weight, thorax width and wing length of 205 *Bombus impatiens* workers captured from 80 colonies placed on 20 sites during the summer of 2016 in Southern Québec, Canada.**

## Pollen

Providing sufficient abundance, pollen diversity is beneficial to bees because it allows them to select flowers according to their nutritional needs [40,41]. In fact, bumble bees are known to show flower constancy in presence of a generous supply of flowers [48,104], a high level of flower richness [48], and a short travel distance to or among flowers or groups of flowers [50]. We hence expected greater flower constancy from workers and a resulting decrease in flower species richness of their pollen loads with more abundant floral resources, especially when local. However, flowering crops did not affect flower constancy. Moreover, intensive crops had barely any effect on the level of flower constancy when local floral resources were species-rich but provoked an increase in flower constancy that became more important as local floral resources became species-poor (Fig 3). Although flower constancy is advantageous for plants because it increases their chances of passing along their pollen grains to a nearby plant of the same species, the benefits of flower constancy for bees are still unclear. Two non-mutually exclusive hypotheses, the "learning" and the "memory" hypotheses, have been put forward to address such benefits. The first implies that pollinators require a substantial amount of time to learn to recognize and handle different types of pollen, while the second suggests that pollinators are limited in their ability to recognize or manipulate more than a few flower types [104]. Either way, it seems more adaptive for workers to focus on a single or very few plant species in presence of sufficient appropriate food resources as they can then become more efficient at finding and extracting pollen and nectar from the plants [105,106]. Our results suggest that such learning and memory constraints may have been exacerbated by the presence of the neonicotinoid pesticides [94,107] associated to intensive row crops in our study area. Note that

our results do not support the possibility that workers switched their pollen selection towards row crops as we did not find significant quantities of cereal pollen in their pollen loads (S2 Table). Future studies will thus face the challenge of not only considering local floral richness as we did, but also attempt to take into account the abundance, quality and pesticide contamination of flowers at various spatial scales.

We expected that the difference in flower species composition of pollen loads and the surrounding local plant community would decrease with (overall) floral resource availability, and especially when flowers are abundant locally. Although there was no effect of flowering crops, our results are in line with this prediction as we found a positive influence of intensive crops on species composition differences that increased in magnitude as flowering crop cover decreased (Fig 3). When flowers were abundant within 500 m, bumble bee workers likely tended to forage closer to the nest [44] and may have thereby reduced the differences in species composition between their pollen loads and locally available plants. In contrast, when surrounded by low proportions of flowering crops, workers probably had to forage further, especially at high intensive crop covers, and thus may have been more likely to encounter or more constrained (see previous paragraph) to use a larger array of plant species. Differences in species composition of pollen loads and surrounding plant communities have mostly been investigated to estimate the distance reached by foraging bees [108] or to evaluate pollen selection, but at the colony level [109,110]. To our knowledge, only one other study compared the plant species composition of pollen loads at the worker level to that of their local plant communities but did not address the understudied determinants of such differences [42].

Lastly, we found as expected a positive correlation between the flower constancy of a bumble bee worker and its body size. The two functional traits most correlated with flower constancy were thorax (intertegular) width and wing length, which both show an allometric relation with body mass [63,111]. Larger workers, at least in *Bombus terrestris*, are considered better at different tasks, including foraging [47,53,55]. Furthermore, larger workers have bigger eyes, which makes them better at detecting flowers and foraging under lower light intensity [112,113]. Hence, larger workers may be able to discriminate and find specific flowers more easily, which thereby allows them to be more flower constant. In spite of this apparent advantage, producing larger workers is not without costs to a colony. For instance, although the largest *B. terrestris* workers are better at gathering resources, their life expectancy and rearing costs are such that the best trade-off for a colony lays with intermediate-size workers [56]. Similarly, intermediate-size *Megachile rotundata* females were found to carry the largest pollen loads [68]. There is clearly a complex trade-off involving the number and size of workers as well as their foraging efficiency, survival and production costs. It would be interesting to determine the role of flower constancy in this trade-off and investigate how landscape structure may modulate the optimal solution(s).

In conclusion, we aimed to investigate the importance of landscape context and local floral resources for bumble bee foraging workers. We found no evidence that worker morphology or size was influenced, likely as a result of low genetic variance or plasticity associated with their commercial rearing, but we observed that their foraging behavior, both in terms of activity and pollen collection, was affected by landscape composition and local floral resources. We also found that bigger workers collected fewer pollen types, which could be an indication of size-dependent foraging behavior or efficiency. It would be interesting to redo the experiment, but with natural colonies to ensure genetic diversity and thus, variability. Furthermore, we bring additional evidence that conservation actions concerning pollinators in agricultural contexts should target landscape-level objectives rather than limit themselves to the farm level.

## Supporting information

**S1 Fig. Habitat composition within 500 m of the 20 clusters (quads) of experimental *Bombus impatiens* colonies monitored in 2016 in Southern Québec, Canada.**
(TIF)

**S2 Fig. Correlation matrix between all landscape explanatory variables from the 20 sites visited during the summer of 2016 in Québec, Canada.**
(TIF)

**S1 Table. ΔAICc comparing both scale (500m vs 1000m) for all models.**
(DOCX)

**S2 Table. List of the plants found in the pollen loads of workers.** # workers represent the number of workers carrying this pollen (n = 264) and the average relative abundance, in percentage.
(DOCX)

## Acknowledgments

We would like to thank all growers who participated in the study; François Gendron and Michelle Breton for their assistance with site selection; Karen Grislis for her help with editing, as well as Guillaume Larocque for statistical advice and Mélissa Girard for pollen identification. We are also grateful to all the summer students, and interns who worked on the project: Audrey Boivin, Damien LeBotland, Thibault Meunier, Benjamin Tremblay-Bergeron, and Olivier Slupik. Finally, we would like to thank Heather Grab as well as two anonymous reviewers for their generous and insightful comments on the manuscript.

## Author Contributions

**Conceptualization:** Amélie Gervais, Valérie Fournier, Marc Bélisle.

**Data curation:** Amélie Gervais, Ève Courtois, Marc Bélisle.

**Formal analysis:** Amélie Gervais, Marc Bélisle.

**Funding acquisition:** Valérie Fournier, Marc Bélisle.

**Investigation:** Amélie Gervais, Ève Courtois, Valérie Fournier, Marc Bélisle.

**Methodology:** Ève Courtois, Valérie Fournier, Marc Bélisle.

**Project administration:** Amélie Gervais, Valérie Fournier, Marc Bélisle.

**Resources:** Ève Courtois, Valérie Fournier.

**Supervision:** Valérie Fournier, Marc Bélisle.

**Visualization:** Amélie Gervais, Marc Bélisle.

**Writing – original draft:** Amélie Gervais.

**Writing – review & editing:** Ève Courtois, Valérie Fournier, Marc Bélisle.

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
