## [Decision Letter · Decision Letter 0]

2 Jan 2020

PONE-D-19-31905

Landscape composition and local floral resources influence foraging behavior but not the size of Bombus impatiens Cresson (Hymenoptera: Apidae) workers

PLOS ONE

Dear Gervais,

Thank you for submitting your manuscript to PLOS ONE. After careful consideration, we feel that it has merit but does not fully meet PLOS ONE’s publication criteria as it currently stands. Therefore, we invite you to submit a revised version of the manuscript that addresses the points raised during the review process.

We would appreciate receiving your revised manuscript by Feb 16 2020 11:59PM. To enhance the reproducibility of your results, we recommend that if applicable you deposit your laboratory protocols in protocols.io, where a protocol can be assigned its own identifier (DOI) such that it can be cited independently in the future. For instructions see: http://journals.plos.org/plosone/s/submission-guidelines#loc-laboratory-protocols

We look forward to receiving your revised manuscript.

Kind regards,

Juliana Hipólito, Phd

Academic Editor

PLOS ONE

Journal Requirements:

"N/A"

Reviewers' comments:

Reviewer's Responses to Questions

**Comments to the Author**

1. Is the manuscript technically sound, and do the data support the conclusions?

Reviewer #1: Partly

Reviewer #2: Yes

2. Has the statistical analysis been performed appropriately and rigorously? 

Reviewer #1: Yes

Reviewer #2: Yes

3. Have the authors made all data underlying the findings in their manuscript fully available?

Reviewer #1: Yes

Reviewer #2: Yes

4. Is the manuscript presented in an intelligible fashion and written in standard English?

Reviewer #1: Yes

Reviewer #2: Yes

5. Review Comments to the Author

Reviewer #1: See below my comment to the editor.

Lorem ipsum dolor sit amet, consectetur adipiscing elit, sed do eiusmod tempor incididunt ut labore et dolore magna aliqua. Ut enim ad minim veniam, quis nostrud exercitation ullamco laboris nisi ut aliquip ex ea commodo consequat. Duis aute irure dolor in reprehenderit in voluptate velit esse cillum dolore eu fugiat nulla pariatur. Excepteur sint occaecat cupidatat non proident, sunt in culpa qui officia deserunt mollit anim id est laborum.

Reviewer #2: Overall, I found the study interesting that there was a negative correlation between the size of workers and the diversity of pollen loads. Most of my comments are small revisions clarifying statements and phrases

Line 18 -19: Unclear why high and low relative covers of intensive and flowering crops would produce smaller workers. Are you hypothesizing that medium cover would produce the larger bees?

Line 20: Awkward phrasing “smaller workers, which together would perform…”, are you saying the workers together would perform. I would say, “smaller workers, who would perform …”

Line 24: Eight workers per colony and week sounds like you collected 8 workers a week from each colony. Rephrase to match your methods.

Lines 62 – 75: The paragraph is awkward. The first sentence makes of this paragraph sets up an argument that there is conflicting information; however it’s not clear to me where the conflict is. The effects of landscape configuration you list are different independent factors on the landscape that influence bee survival, as you mention in the second to last sentence of the paragraph. I’m not sure where the inconsistencies you allude to. I would remove the yet in line 64 as that sentence does not contradict the previous sentence. I would also remove the word definitely from line 74 as it sounds flippant.

Line 76: I think the sentence would read better without “yet.”

Line 81. You say “yet, again,” but you have not raised this issue before so I am not sure why you say yet, again. I would also say specifically that you think the individual workers are the mechanism by which landscape characteristics influence bumble bee health.

Line 85: Not sure what Phacelia Jussieu is referring to, but the species name was Phacelia tranacetifolia

Line 86: I would say foraging trips instead of provisioning to be clearer since provisioning also refers to the act of nurse bees feeding the broad.

Line 116 – 117: You talk about efficiency in collecting pollen, but you did not measure efficiency, you measured the consistency of pollen. I’m not sure how you can make the leap between pollen load homogenization and efficiency. A worker can be very slow at collecting pollen and still be have a homogenous pollen load. Also, why do you expect larger workers to be more constant? This hypothesis is introduced in this sentence but never referred to again.

Line 143: Was the time standardized when foraging activity was measured.

Line 178: Clarify what you mean by each land use type. What is the difference between intensive and extensive crops?

Line 188: You considered habitats within 1000-m of each colony because bumble bees can forage that far but you only identified blooming plants within a 100-m radius. It seems like you could be missing a large portion of the flowers that these bees are foraging on. Was there a reason for this difference?

Line 207: It is unfortunate that you could not include the proportion of forest and total margin length in the model.

Line 217: I would remove “pushed the envelope further.” This is too casual and I’m not clear why this is “pushing the envelope.”

Line 251: Why were 59 workers excluded from the study?

Line 254: I would include the caveat that foragers showed little variation. There still could differences in variation if you included the in-nest workers.

Line 269: Again, I would say foraging activity instead of food-provisioning to be clearer since nurse workers provision the brood.

Line 294 – 297; Since all of the morphological traits are correlated, I would report only one trait that correlated with pollen load species richness. Alternatively, present all of them, just don’t present a subset of them because it is not clear why these where cherry-picked to be presented.

Line 331: It is unclear what you mean by “including ours” as you did not measure any morphological measurements correlated with intensive crops. Do you mean previous studies in the same environment?

Line 343: Again, unclear what you mean by “including ours”

Line 357: I would write “barely any effect” instead of “barely no effect.”

Line 367: Could they just be switching to the intensive crop for pollen and that is why richness drops.

Line 369: Remove “to push the envelope further,” to casual and unclear why this is “pushing the envelope.”

Line 401: I’m not sure why you say you found that worker morphology or size was not influenced when you found that size decreased with increasing floral diversity. I would include this result in your conclusion.

Figure 3: Include data points on the graph

Figure 4: The legend has a line and a dashed line but the graph only has two solid lines.

Figure 5: What do the two lines represent?

6. PLOS authors have the option to publish the peer review history of their article (what does this mean?). If published, this will include your full peer review and any attached files.

Reviewer #1: No

Reviewer #2: No

---

## [Author Response · Author response to Decision Letter 0]

3 Mar 2020

My co-authors and I are really grateful for the opportunity to resubmit our manuscript (PONE-D-19-31905) entitled “Landscape composition and local floral resources influence foraging behavior but not the size of Bombus impatiens Cresson (Hymenoptera: Apidae) workers” for consideration for publication by Plos One. We would like to thank the reviewer for the excellent comments we received. All comments/suggestions made by the reviewer have been addressed, with corresponding changes made directly to the manuscript. This letter is accompanied by a revised version of our manuscript as well as its corresponding document in track-change format. Detailed responses to comments or concerns can be found in the pages below. 

Best regards, 

Amélie Gervais

PhD candidate  

Reviewer #2: Line 18 -19: Unclear why high and low relative covers of intensive and flowering crops would produce smaller workers. Are you hypothesizing that medium cover would produce the larger bees?

Authors: We agree with the reviewer that the sentence was confusing. We accordingly changed the phrasing. Lines: 22-25

Reviewer #2: Line 20: Awkward phrasing “smaller workers, which together would perform…”, are you saying the workers together would perform. I would say, “smaller workers, who would perform …”

Authors: Modified as suggested. Line: 24

Reviewer #2: Line 24: Eight workers per colony and week sounds like you collected 8 workers a week from each colony. Rephrase to match your methods.

Authors: Done. Lines: 27-29

Reviewer #2: Lines 62 – 75: The paragraph is awkward. The first sentence makes of this paragraph sets up an argument that there is conflicting information; however it’s not clear to me where the conflict is. The effects of landscape configuration you list are different independent factors on the landscape that influence bee survival, as you mention in the second to last sentence of the paragraph. I’m not sure where the inconsistencies you allude to.

Authors: We wanted to put emphasis on the fact that it is difficult to generalize conclusions obtained through different studies that addressed the influence of landscape configuration on bumble bees. For instance, different metrics are used for characterizing landscape configuration, and, even though metrics assessing a specific aspect of configuration are theoretically correlated, results often differ according to the metrics used. We modified the paragraph to make that thought clearer. Lines: 66-80

I would remove the yet in line 64 as that sentence does not contradict the previous sentence. 

Authors: Done. Line: 81

I would also remove the word definitely from line 74 as it sounds flippant.

Authors: Done. Lines: 78-79

Reviewer #2: Line 76: I think the sentence would read better without “yet.”

Authors: Done. Line: 81

Reviewer #2: Line 81. You say “yet, again,” but you have not raised this issue before so I am not sure why you say yet, again. I would also say specifically that you think the individual workers are the mechanism by which landscape characteristics influence bumble bee health.

Authors: We agree with the reviewer and accordingly changed the wording to improve clarity. Line: 86

Reviewer #2: Line 85: Not sure what Phacelia Jussieu is referring to, but the species name was Phacelia tranacetifolia

Authors: Jussieu is the authority of the genus, but we added the species name to be clearer. Line: 90

Reviewer #2: Line 86: I would say foraging trips instead of provisioning to be clearer since provisioning also refers to the act of nurse bees feeding the broad.

Authors: Done. Line: 91 

Reviewer #2: Line 116 – 117: You talk about efficiency in collecting pollen, but you did not measure efficiency, you measured the consistency of pollen. I’m not sure how you can make the leap between pollen load homogenization and efficiency. A worker can be very slow at collecting pollen and still be have a homogenous pollen load. Also, why do you expect larger workers to be more constant? This hypothesis is introduced in this sentence but never referred to again.

Authors: We thought that flower constant workers would be more efficient in collecting pollen because they would learn how to handle the flowers of a few plants and thereby get really good at it. We hence thought that flower constancy would lead to foraging efficiency. But, as you pointed out, we have data on pollen loads but not on the ‘’time’’ component needed to assess efficiency. We changed our wording and now simply refer to pollen consistency as suggested. Lines: 121-122

As for our expectation that larger workers would be more flower constant, we said in the previous paragraph (Lines:102-103) that bigger workers are usually better foragers. We expect that a better forager would choose to be flower constant whenever possible. 

Reviewer #2: Line 143: Was the time standardized when foraging activity was measured.

Authors: We counted workers entries and exits from each colony for 15 minutes. As for the statistical analysis, we considered the number of workers/15 minutes. Furthermore, counts were made between 9h00 and 16h00 and colonies were visited at different times across weeks to avoid confounding time effects. (Lines:147-148)

Reviewer #2: Line 178: Clarify what you mean by each land use type. What is the difference between intensive and extensive crops?

Authors: Precisions were added within the text. (Lines: 185-187) 

Reviewer #2: Line 188: You considered habitats within 1000-m of each colony because bumble bees can forage that far but you only identified blooming plants within a 100-m radius. It seems like you could be missing a large portion of the flowers that these bees are foraging on. Was there a reason for this difference?

Authors: It was not logistically possible to identify every plant within a 1000-m radius from the colonies. We did not have either the time or financial / human resources associated with that level of precision. Habitats were easier and faster to characterize, because a field or a forest could cover quite a large proportion of the radius. It might have been possible using drones, but we did not have the equipment, experience or financial resources to use that kind of tool in 2016.

Reviewer #2: Line 207: It is unfortunate that you could not include the proportion of forest and total margin length in the model.

Authors: Yes, we were really disappointed too. 

Reviewer #2: Line 217: I would remove “pushed the envelope further.” This is too casual and I’m not clear why this is “pushing the envelope.”

Authors: We removed the expression as suggested. Line: 226 

Reviewer #2: Line 251: Why were 59 workers excluded from the study?

Authors: They were excluded because they had missing body parts. This was already mentioned on Lines: 168-169

Reviewer #2: Line 254: I would include the caveat that foragers showed little variation. There still could differences in variation if you included the in-nest workers.

Authors: Done. Lines: 266

Reviewer #2: Line 269: Again, I would say foraging activity instead of food-provisioning to be clearer since nurse workers provision the brood.

Authors: Done. Lines: 292

Reviewer #2: Line 294 – 297; Since all of the morphological traits are correlated, I would report only one trait that correlated with pollen load species richness. Alternatively, present all of them, just don’t present a subset of them because it is not clear why these where cherry-picked to be presented.

Authors: We decided to present all of them instead. Lines: 337-339

Reviewer #2: Line 331: It is unclear what you mean by “including ours” as you did not measure any morphological measurements correlated with intensive crops. Do you mean previous studies in the same environment?

Authors: It was effectively not clear, so we removed the expression.

Reviewer #2: Line 343: Again, unclear what you mean by “including ours”

Authors: It was effectively not clear, so we removed the expression. 

Reviewer #2: Line 357: I would write “barely any effect” instead of “barely no effect.”

Authors: Done as suggested. Line: 402 

Reviewer #2: Line 367: Could they just be switching to the intensive crop for pollen and that is why richness drops.

Authors: Poaceae pollen (mostly associated to intensive row crops) was often found, but never in high abundance. It looked more like accidental carrying than purposely selected pollen (~3%) as Poaceae pollen is usually carried by wind. We added a column in the supplementary Table 2 representing the averaged relative abundance of pollen (%) transported by bees for each plant. We also added a sentence in the text to signify that we thought about that possibility. Lines: 414-416

Reviewer #2: Line 369: Remove “to push the envelope further,” to casual and unclear why this is “pushing the envelope.”

Authors: The expression was removed as suggested. 

Reviewer #2: Line 401: I’m not sure why you say you found that worker morphology or size was not influenced when you found that size decreased with increasing floral diversity. I would include this result in your conclusion.

Authors: Thank you for pointing that out. In fact, we found no variation of body size with landscape habitat composition, but a negative correlation between body size and the species richness of pollen loads. We added this result in the conclusion. Lines: 451-452

Reviewer #2: Figure 3: Include data points on the graph

Authors: Done. See Figure 3. 

Reviewer #2: Figure 4: The legend has a line and a dashed line but the graph only has two solid lines.

Authors: This has to be a mistake with the pdf, because we really have one dashed line, and a solid line. We will make sure that the dashed line is visible. 

Reviewer #2: Figure 5: What do the two lines represent?

Authors: Thank you for pointing out this oversight. We added a legend in the figure.

---

## [Decision Letter · Decision Letter 1]

7 Apr 2020

PONE-D-19-31905R1

Landscape composition and local floral resources influence foraging behavior but not the size of Bombus impatiens Cresson (Hymenoptera: Apidae) workers

PLOS ONE

Dear Gervais,

Thank you for submitting your manuscript to PLOS ONE. After careful consideration, we feel that it has merit but does not fully meet PLOS ONE’s publication criteria as it currently stands. Therefore, we invite you to submit a revised version of the manuscript that addresses the points raised during the review process.

Dear authors, 

I hope you all are ok considering our world situation. We are having a hard time to find available reviewers for publication, covid-19 make it harder. 

This time I recieved two evaluations for your manuscript. One of then still suggests some major points that deserves a second look on the paper. I'm certain that this changes will improve it. 

We would appreciate receiving your revised manuscript by May 22 2020 11:59PM. To enhance the reproducibility of your results, we recommend that if applicable you deposit your laboratory protocols in protocols.io, where a protocol can be assigned its own identifier (DOI) such that it can be cited independently in the future. For instructions see: http://journals.plos.org/plosone/s/submission-guidelines#loc-laboratory-protocols

We look forward to receiving your revised manuscript.

Kind regards,

Juliana Hipólito, Phd

Academic Editor

PLOS ONE

Reviewers' comments:

Reviewer's Responses to Questions

**Comments to the Author**

1. If the authors have adequately addressed your comments raised in a previous round of review and you feel that this manuscript is now acceptable for publication, you may indicate that here to bypass the “Comments to the Author” section, enter your conflict of interest statement in the “Confidential to Editor” section, and submit your "Accept" recommendation.

Reviewer #2: All comments have been addressed

Reviewer #3: (No Response)

2. Is the manuscript technically sound, and do the data support the conclusions?

Reviewer #2: Yes

Reviewer #3: No

3. Has the statistical analysis been performed appropriately and rigorously? 

Reviewer #2: Yes

Reviewer #3: No

4. Have the authors made all data underlying the findings in their manuscript fully available?

Reviewer #2: Yes

Reviewer #3: No

5. Is the manuscript presented in an intelligible fashion and written in standard English?

Reviewer #2: Yes

Reviewer #3: Yes

6. Review Comments to the Author

Reviewer #2: The authors have sufficiently addressed all of my previous comments.

Reviewer #3: Reviewer: Heather Grab

First, I hope that both the journal staff, authors and their families are safe as we all face the challenges of covid-19.

The manuscript presents results of a large field study which evaluates the impacts of local and landscape-scale resources on individual forager traits and on colony-level foraging activity. I was very excited to see this study as my own research is at the intersection of intraspecific morphological variation and landscape ecology. The authors have done a good job addressing previous reviewer comments; however, I have some additional points that the authors should address before publication.

***Please see attached reviewer file for comments along with inline code***

Major comments:

I found it very difficult to interpret the results from the model comparisons reported by the authors. As I understand, the authors conducted comparisons based on a pre-defined set of candidate models. They explained why some landcovers (border, urban, forest) were excluded from their models, however others (extensive covers) seem to be left out without justification even though they comprise a substantial portion (~50%) of the landscape at some sites. Table 2 reports Model-averaged coefficients but which model sets were used to obtain these averages is unclear as they appear to be reported on an individual model basis? In table three, I suggest labeling each candidate model by its predictors as in Table 1, because it is currently very difficult to understand the differences between models.

Given these difficulties, I was very happy to see that the author’s provided a Dryad link to the underlying data supporting their analysis. However, when looking at the data I noticed a few issues that the authors can hopefully resolve:

First, the authors report in their methods that foraging bees were only collected starting four weeks after the colonies were set out at each site (May 3). This makes a lot of sense, as the goal of the study was to measure variation in workers relative the resources available at a sites, and therefore the authors would want to be sure that they were measuring workers actually provisioned at the site rather than those that came with the colony or were provisioned primarily with the nectar and pollen resources provided to the colony during factory rearing. However, in looking at the worker datafile, many workers appear to be collected before this 4 week period (May 18-31st). After excluding these data, I no longer see a significant effect of JJ (Julian date) on the morphological measures.

The effect of date is no longer significant after accounting for the 4 week window. This suggest that the effect was largely due to workers produced before the start of the study and that worker size is smaller once colonies are placed in the field.

In running some additional models with the Extensive cover variable on the reduced dataset, I found that an interaction with local flower cover did explain significant variation in several of the morphological traits (thorax shown here as an example). Using the lme4 and lmerTest packages I was able to account for the nested random effect of Hive within Site without running into model singularity or convergence issues. As the majority of variables considered by the author are continuous and have this hierarchical structure, I recommend using this package rather the glmmTMB. Additionally, the data provided by the authors appears to only have the morphological measures for the bees which had weight data taken (205 rather than the full 264 reported in the results). Because this is actually quite a small sample size given the date range, number of colonies and sites sampled, I evaluated whether there were sites with only a few bees collected (<5) and removed them from the analysis. This ensures that results describe the patterns only for sites with a sample sufficient to accurately characterize the average size of workers within a site.

These results suggest that then local floral resources are high, worker weight declines with greater ag cover but that ag cover can have the opposite effect, increasing body size when local floral resources are low.

I am also concerned with the use of a correlation test when evaluating whether body size influences the number of pollen types carried by foragers. First because foragers are not all independent (multiple foragers from the same sites) but more importantly because local floral resources vary across sites, and if above analyses hold true, then variation in floral resources is related to body size in at least some landscape conditions. Because pollen richness is count data, I recommend a poisson glm model.

Proposed model structure for pollen richness vs body size analysis. I used bees from all dates and did not set a minimum replication threshold.

m=glmer(Pollen_load_richness~LOC_10w+scale(thorax_avg)+(1|Site/Hive), data=subset(dat, ), family="poisson")

summary(m)

## Generalized linear mixed model fit by maximum likelihood (Laplace

## Approximation) [glmerMod]

## Family: poisson ( log )

## Formula: Pollen_load_richness ~ LOC_10w + scale(thorax_avg) + (1 | Site/Hive)

## Data: subset(dat, )

##

## AIC BIC logLik deviance df.resid

## 1170.1 1186.7 -580.0 1160.1 200

##

## Scaled residuals:

## Min 1Q Median 3Q Max

## -2.4051 -0.9661 -0.2117 0.7685 5.4298

##

## Random effects:

## Groups Name Variance Std.Dev.

## Hive:Site (Intercept) 0.10302 0.3210

## Site (Intercept) 0.01575 0.1255

## Number of obs: 205, groups: Hive:Site, 76; Site, 20

##

## Fixed effects:

## Estimate Std. Error z value Pr(>|z|)

## (Intercept) 1.45164 0.18518 7.839 4.54e-15 ***

## LOC_10w 0.03019 0.02202 1.371 0.17037

## scale(thorax_avg) -0.10057 0.03242 -3.102 0.00192 **

## ---

## Signif. codes: 0 '***' 0.001 '**' 0.01 '*' 0.05 '.' 0.1 ' ' 1

##

## Correlation of Fixed Effects:

## (Intr) LOC_10

## LOC_10w -0.952

## scl(thrx_v) 0.012 0.004

Results indicate that larger individuals still collect fewer pollen types even after accounting for repeated measures within sites and for site level variation in floral richness

The authors should certainly re-run each of these models with their full dataset. Because the authors did not provide the foraging activity data, I would encourage them to reassess these models as well.

L311-312: while the authors are correct that pollen load richness declines increasing proportions of intensive crop cover when local floral richness is low it does not follow from the data to suggest that when local floral richness is high, pollen load richness increases with intensive crop cover. The pattern in figure 3 suggests that this relationship is likely not different than zero.

Minor comments:

L67-83 discussion of how differences in metric limits synthesis seems outside of the scope of the current work given that the authors are not attempting a synthesis effort. I would limit this section only to explaining why particular metrics might be expects to impact body size and foraging variation.

L130-131: It would be very helpful if the authors would provide a map of their study sites, based on their regional description it seems that spatial autocorrelation many be an issue. Given this concern, I suggest that the authors confirm that model residuals do not display spatial autocorrelation.

7. PLOS authors have the option to publish the peer review history of their article (what does this mean?). If published, this will include your full peer review and any attached files.

Reviewer #2: No

Reviewer #3: Yes: Heather Grab

---

## [Author Response · Author response to Decision Letter 1]

22 May 2020

22- May- 2020

Dear Dr. Hipólito

My co-authors and I are really grateful for the opportunity to resubmit our manuscript (PONE-D-19-31905) entitled “Landscape composition and local floral resources influence foraging behavior but not the size of Bombus impatiens Cresson (Hymenoptera: Apidae) workers” for consideration for publication by Plos One. We would like to thank the reviewer for the relevant comments we received. All comments/suggestions made by the reviewer have been addressed, with corresponding changes made directly to the manuscript. This letter is accompanied by a revised version of our manuscript as well as its corresponding document in track-change format. Detailed responses to comments or concerns can be found in the pages below. 

Best regards, 

Amélie Gervais, PhD

Reviewer: First, I hope that both the journal staff, authors and their families are safe as we all face the challenges of covid-19. 

Answer: My coauthors and I are all well, thank you. We hope that you are too. 

Reviewer: The manuscript presents results of a large field study which evaluates the impacts of local and landscape-scale resources on individual forager traits and on colony-level foraging activity. I was very excited to see this study as my own research is at the intersection of intraspecific morphological variation and landscape ecology. The authors have done a good job addressing previous reviewer comments; however, I have some additional points that the authors should address before publication. 

Answer: Thank you for your comments.

Reviewer: I found it very difficult to interpret the results from the model comparisons reported by the authors. As I understand, the authors conducted comparisons based on a pre-defined set of candidate models. They explained why some landcovers (border, urban, forest) were excluded from their models, however others (extensive covers) seem to be left out without justification even though they comprise a substantial portion (~50%) of the landscape at some sites. 

Answer: We decided to exclude the relative amount of extensive cultures composed of hayfields and pastures, because we considered this landcover type to be more ‘’neutral’’ than others with respect to bumble bees. Indeed, while hayfields and pastures are rarely treated with pesticides, they are also generally characterized by a much lower food resource availability to bumble bees than alfalfa and clover fields, at least in our study area. We agree that it would have been nice to consider the influence of a greater number of landcovers in our regression models, yet we believe that this may lead to another problem as landcover variables are “compositional” variables, that is that they sum to a constant and thus imply a lack of independence among them (i.e., they are implicitly correlated; Filzmoser et al. 2009). This may not only lead to collinearity problems within a given model, but also to interpretation problems when contrasting models. Given the above, and the fact that we do not want to propose post hoc hypotheses or predictions regarding extensive covers that could turn out to be “false positives” (Forstmeier et al. 2017), we would prefer to keep the composition of our model sets intact. This being said, we now clarify why we kept the relative amount of extensive cultures out of our analyses. Lines: 217-219 and 221-226. 

Filzmoser, P., K. Hron, and C. Reimann. 2009. Principal component analysis for compositional data with outliers. Environmetrics 20:621-632.

Forstmeier, W., E.-J. Wagenmakers, and T. H. Parker. 2017. Detecting and avoiding likely false-positive findings – a practical guide. Biological Reviews 92:1941-1968.

Reviewer: Table 2 reports Model-averaged coefficients but which model sets were used to obtain these averages is unclear as they appear to be reported on an individual model basis? 

Answer: We now provide in Table 2 the model numbers that were used to compute the model-averaged coefficients. Line: 307.

Reviewer: In table three, I suggest labeling each candidate model by its predictors as in Table 1, because it is currently very difficult to understand the differences between models.

Answer: We decided to combine Table 1 and 3 for more clarity. Line: 245

Reviewer: Given these difficulties, I was very happy to see that the author’s provided a Dryad link to the underlying data supporting their analysis. However, when looking at the data I noticed a few issues that the authors can hopefully resolve:

First, the authors report in their methods that foraging bees were only collected starting four weeks after the colonies were set out at each site (May 3). This makes a lot of sense, as the goal of the study was to measure variation in workers relative the resources available at a sites, and therefore the authors would want to be sure that they were measuring workers actually provisioned at the site rather than those that came with the colony or were provisioned primarily with the nectar and pollen resources provided to the colony during factory rearing. However, in looking at the worker datafile, many workers appear to be collected before this 4 week period (May 18-31st). After excluding these data, I no longer see a significant effect of JJ (Julian date) on the morphological measures. 

Answer: Thank you for pointing that out! In fact, we started to collect workers two weeks after placing quads in the field in order to sample pollen loads, but wanted to exclude the workers caught between the second and fourth week of the experiment to avoid measuring effects biased by initial conditions. We clarified this in the manuscript and all morphological analyses were rerun using the correct subset of data instead of all the workers. These changes barely changed the results except for Julian date which had no effect in the new analyses (as found by the reviewer). Lines: 148 and 212-215. 

Reviewer: In running some additional models with the Extensive cover variable on the reduced dataset, I found that an interaction with local flower cover did explain significant variation in several of the morphological traits (thorax shown here as an example). Using the lme4 and lmerTest packages I was able to account for the nested random effect of Hive within Site without running into model singularity or convergence issues. As the majority of variables considered by the author are continuous and have this hierarchical structure, I recommend using this package rather the glmmTMB. Additionally, the data provided by the authors appears to only have the morphological measures for the bees which had weight data taken (205 rather than the full 264 reported in the results). Because this is actually quite a small sample size given the date range, number of colonies and sites sampled, I evaluated whether there were sites with only a few bees collected (<5) and removed them from the analysis. This ensures that results describe the patterns only for sites with a sample sufficient to accurately characterize the average size of workers within a site. 

Answer: We did not feel comfortable with considering the influence of extensive crops post-hoc, and this, for the reasons mentioned above. This being said, we totally understand the concerns of the reviewer and we reran all our analyses with the largest sample size possible for each trait with the additional constraint that sites had to include at least 5 workers as suggested. We also used the lme4 package for these analyses as suggested by the reviewer (except for the foraging activity, where we still faced convergence issues for some models with lme4, but not with glmmTMB). These modifications did not affect our results nor our conclusions. Finally, we added the foraging activity data and the workers that were not weighted due to missing body parts to the Dryad dataset as requested by the reviewer. Lines: 258-260. 

Reviewer: I am also concerned with the use of a correlation test when evaluating whether body size influences the number of pollen types carried by foragers. First because foragers are not all independent (multiple foragers from the same sites) but more importantly because local floral resources vary across sites, and if above analyses hold true, then variation in floral resources is related to body size in at least some landscape conditions. Because pollen richness is count data, I recommend a poisson glm model. 

 Answer: Given that there are no way to compute a linear correlation coefficient between two variables subjected to a hierarchical structure, we nevertheless agree with the reviewer that her approach is likely better, inasmuch as it allows to control for the influence of local floral richness. We therefore conducted the analyses as suggested. This being said, the new results are in line with those of our previous analyses and did not affect our conclusions. Lines: 280-284 and 354-359. 

Reviewer: The authors should certainly re-run each of these models with their full dataset. Because the authors did not provide the foraging activity data, I would encourage them to reassess these models as well. 

Answer: We reran all the models as suggested. The changes can be observed in Table 2 and Table 3, as well as in Fig. 1, but these brought no qualitative modifications to the relationship that we had found. 

Reviewer: L311-312: while the authors are correct that pollen load richness declines increasing proportions of intensive crop cover when local floral richness is low it does not follow from the data to suggest that when local floral richness is high, pollen load richness increases with intensive crop cover. The pattern in figure 3 suggests that this relationship is likely not different than zero. 

Answer: Thanks for pointing that out. We had mistaken flower constancy for pollen load richness, which are inversely proportionate. We changed the phrasing in both the results and the discussion. Lines: 329-332 and 423-424. 

Reviewer: L67-83 discussion of how differences in metric limits synthesis seems outside of the scope of the current work given that the authors are not attempting a synthesis effort. I would limit this section only to explaining why particular metrics might be expects to impact body size and foraging variation. 

Answer: While we reduced this section, we believe it is important to provide the reader with a good background regarding the difficulty in making predictions about the effects of landscape structure on bumble bee morphology or behavior. Lines: 66-67, 78. 

Reviewer: L130-131: It would be very helpful if the authors would provide a map of their study sites, based on their regional description it seems that spatial autocorrelation many be an issue. Given this concern, I suggest that the authors confirm that model residuals do not display spatial autocorrelation. 

Answer: We now provide a map of our study sites (S1 Fig). Spatial autocorrelation of residuals was assessed for each response variable using Moran’s I and the most complex model via the DHARMa package in R. The latitude and longitude of each site were also added in the Dryad dataset. No spatial autocorrelation was found. This is now clearly mentioned in the manuscript. Lines: 241-244.

---

## [Editor Report · Decision Letter 2]

28 May 2020

Landscape composition and local floral resources influence foraging behavior but not the size of Bombus impatiens Cresson (Hymenoptera: Apidae) workers

PONE-D-19-31905R2

Dear Dr. Gervais,

We are pleased to inform you that your manuscript has been judged scientifically suitable for publication and will be formally accepted for publication once it complies with all outstanding technical requirements.

With kind regards,

Juliana Hipólito, Phd

Academic Editor

PLOS ONE

---

## [Editor Report · Acceptance letter]

15 Jun 2020

PONE-D-19-31905R2 

Landscape composition and local floral resources influence foraging behavior but not the size of *Bombus impatiens* Cresson (Hymenoptera: Apidae) workers 

Dear Dr. Gervais:

I'm pleased to inform you that your manuscript has been deemed suitable for publication in PLOS ONE. Congratulations! Your manuscript is now with our production department. 

Kind regards, 

on behalf of

Dr. Juliana Hipólito 

Academic Editor

PLOS ONE